# Prevention and Control of COVID-19 after Resuming General Hospital Functions

**DOI:** 10.3390/pathogens11040452

**Published:** 2022-04-10

**Authors:** Jingwen Li, Hanshu Liu, Chao Duan, Lan Chen, Qing Zhang, Xi Fang, Lei Tan, Na Li, Xinyi Wang, Xing Zhang, Chunmei Li, Zhicheng Lin, Nian Xiong

**Affiliations:** 1Department of Neurology, Union Hospital, Tongji Medical College, Huazhong University of Science and Technology, Wuhan 430000, China; jingwenli1009@163.com (J.L.); liu1749676236@163.com (H.L.); wangxinyi_21@163.com (X.W.); 2Medical Treatment Expert Group for COVID-19, Wuhan Red Cross Hospital, Wuhan 430000, China; ljw18982747407@163.com (C.D.); m201975801@hust.edu.cn (L.C.); zaqi1990@163.com (Q.Z.); hszh18062518996@163.com (X.F.); tl18062518898@163.com (L.T.); nana39160@163.com (N.L.); zx08887@163.com (X.Z.); 3Wuhan Dongxihu District People’s Hospital, Wuhan 430000, China; 4Laboratory of Psychiatric Neurogenomics, McLean Hospital, Harvard Medical School, Belmont, MA 02478, USA

**Keywords:** hospital management, infection control, COVID-19, function, resume

## Abstract

During the COVID-19 pandemic, many general hospitals have been transformed into designated infectious disease care facilities, where a large number of patients with COVID-19 infections have been treated and discharged. With declines in the number of hospitalizations, a major question for our healthcare systems, especially for these designated facilities, is how to safely resume hospital function after these patients have been discharged. Here, we take a designated COVID-19-care facility in Wuhan, China, as an example to share our experience in resuming hospital function while ensuring the safety of patients and medical workers. After more than 1200 patients with COVID-19 infections were discharged in late March, 2020, our hospital resumed function by setting up a three-level hospital infection management system with four grades of risk of exposure. Moreover, we also took measures to ensure the safety of medical personnel in different departments including clinics, wards, and operation rooms. After all patients with COVID-19 infections were discharged, during the five months of regular function from April to September in 2020, no positive cases have been found among more than 40,000 people in our hospital, including hospital staff and patients.

## 1. Introduction

It has been three years since the initial COVID-19 outbreak. A total of 86 hospitals in Wuhan had to be converted into designated hospitals to receive only patients with COVID-19 infections during the pandemic. Many departments, including the stomatology department and the department of plastic surgery, could not treat their patients as usual. Many patients with chronic diseases, such as Parkinson’s disease, could neither be re-examined nor acquire medicine normally, which caused great difficulty for patients, hospitals, and society at large. Previously, we shared our experience on transforming a regularly functioning general hospital into a designated infectious disease care facility for COVID-19 [1]. In late March, 2020, after extensive quarantine, the number of patients with active COVID-19 infections decreased dramatically, so all remaining patients were transferred to just 10 designated hospitals and the other 76 hospitals, including ours, returned to being non-designated, regularly functioning hospitals for normal admission of patients with other medical conditions. After the results of mass testing for SARS-CoV-2 in nucleic acid came back negative in Wuhan City in April, 2020, citizens have been returning to normal life and businesses and industries have been gradually reviving. In order to prevent a rebound of the epidemic, according to the instructions of the Chinese government, hospitals should be carrying out regular prevention and control work. Therefore, how to safely resume hospital function within a short period of time while minimizing the risk of a next “wave” and imported carriers after the pandemic will become an unprecedented issue [2]. Here, we take the experience of China’s epicenter hospital, Wuhan Red Cross Hospital (WRCH), which is only a mile away from Huanan Seafood Wholesale Market, as an example to discuss the safety measures implemented during resumption of regular function (Figure 1). WRCH was one of the first hospitals designated for COVID-19 infections and is the closest to Huanan seafood market, treating more than 1200 COVID-19 patients during a period of three months from January to April 2020. The biggest reason for the difficulty in addressing the epidemic was the lack of awareness and preparation in dealing with this novel virus during the initial outbreak. Therefore, after the reopening of WRCH, we performed a comprehensive, regular mass screening of all ages in nearby districts [3] in order to detect potential patients and to avoid new transmission of the virus. The transmission of SARS-CoV-2 is primarily airborne [4]. Another route of infection via fomites was believed to be significant early in the pandemic but has subsequently been shown to be of minor importance [5]. These principles guided our infection-control procedures.

## 2. Establishing a Three-Level Hospital Infection Management System

The key feature in resuming hospital function is infection prevention. For this purpose, three levels were created within the hospital-wide organizational system. The first level is an infection control committee, composed of the hospital president and leaders of the infection management department, medical department, pharmacy department, nursing department, outpatient office, general affairs department, consumables department, pharmacy department, security department, and other functional departments. The president or vice president is the first person responsible for and in charge of overall command including emergency response, personnel management, material allocation, and logistic support. The second level includes the staff of the hospital infection management department, which is responsible for formulating the systems and processes for infection prevention and control, personnel training, on-site guidance, inspection and supervision of disinfection and isolation, health condition monitoring, occupational exposure monitoring, and material inspection. The third level is a hospital infection management team, consisting of doctors and nurses from all departments. They are responsible for the implementation of infection control measures, health monitoring and reporting, occupational exposure monitoring and reporting, personal protection, environmental cleaning and disinfection, etc. The staff of the hospital infection department are responsible for supervising the outpatient department and making rounds in each ward at least once a month. More importantly, proper and timely communication between different levels is a critical component of the hospital infection management system. Specifically, first-level personnel should hold regular meetings to report on the state of hospital infections and make further recommendations. At the same time, the person in charge should perform his/her duties, including random inspections of key departments such as the fever clinic, isolation ward, and medical technology department; of medical waste disposal; and of air and equipment disinfection. Timely rectification and follow-ups are key to ensuring continuous improvement in COVID-19 prevention and high-quality maintenance.

## 3. Managing Outpatient Department and Wards for Everyone

### 3.1. General Clinic

All consulting rooms are equipped with methanol-free, quick hand disinfectants. In addition, each room is equipped with an ultraviolet lamp or air disinfection machine. The outpatient areas are carefully cleaned, disinfected, and ventilated by a group of trained medical cleaners every day. In addition, we established a procedure for outpatient registration so that patients can make time-sharing appointments to see a doctor, allowing the hospital to avoid having patients gathered in the registration hall and waiting area. All personnel entering the waiting area must wear masks correctly, have a normal body temperature, and provide a “Health Code” (a QR code shown on their phone to indicate their level of health risk. This code also records the person’s travels within the past few days, including transportation information and places visited). In addition, patients and escorts must maintain a safe distance from others while waiting, which is recommend to be six feet or more.

### 3.2. Separating Fever Clinic

The fever clinic in our hospital has a clear logo and is set up separately from the general outpatient and emergency departments. It has an independent entrance to facilitate the transfer of patients. Three areas (clean area, potentially polluted area, and polluted area) and five passageways (patient channel, employee channel, administrative personnel channel, cleaning personnel channel, and sewage channel) are established to meet the requirements of hospital infection prevention. Clinics and isolation observation rooms are each equipped with disinfection facilities and professional medical staff, specifically, one doctor, one nurse, and several cleaners for each room. In addition, all patients routinely undergo the nucleic acid test, antibody test, chest CT, and blood routine examination in order to determine if patients have been infected.

Notably, WRCH formulated an emergency plan for outpatients who need emergency surgery but fail to provide their screening results for COVID-19. Specifically, those patients are allowed to be admitted into the hospital without contact and transferred to a dedicated operation room for timely treatment. In the meantime, detection staff would conduct rapid screening.

## 4. Inpatient Medical Ward

All personnel entering the hospital should wear masks correctly and pay attention to hand hygiene (i.e., wash hands frequently and after every examination). The hospital strictly enforces the requirements for relevant management of the “four categories of personnel” (medical workers, executives, patients, and caregivers). In principle, all of them should report their body temperature to the infection control department daily and check their nucleic acid at regular intervals (e.g., once per week). Patients and caregivers should not leave the ward without permission during hospitalization to avoid cross infection, while medical workers and executives should pay more attention to self-protection and their responsibilities. Moreover, we set up a “buffer ward” system. Patients who need to be hospitalized urgently are first admitted to the buffer ward and then transferred to the general ward after screening. Specifically, the purpose of a “buffer ward” is to reduce the pressure on other wards and to provide sufficient time to screen and classify patients. If patients who are hospitalized have a fever, they are immediately transferred to the isolation ward. Before using an air conditioner, good ventilation must be maintained in the room for 20–30 min. After any patients with COVID-19 infections were discharge and before returning to normal functioning as a general hospital, all air conditioning and ventilation systems were disinfected and cleaned completely. The hospital did not reopen until it passed hygiene inspection and evaluation by the CDC.

## 5. Overseeing Healthcare versus Visitation

Our general principle is “one patient, one caregiver” with fewer face-to-face but more video visits. Caregivers may only enter the wards with an official card after obtaining the results of their routine examinations including SARS-CoV-2 nucleic acid and antibody tests, chest CT examination, and routine blood analysis. Caregivers were instructed to wear masks every day and to wash their hands frequently. Additionally, they were responsible for ensuring good personal protection and taking their temperature daily to prevent cross infection. In general, three meals per day were provided by the hospital canteen on time, reducing contact with people outside the hospital.

## 6. Keeping Operation Rooms Disinfected

For either emergency or non-emergency medical operations, the interval taken for disinfections between two operations should not be less than one hour [6]. Patients undergoing non-emergency surgery are expected to complete preoperative screening for COVID-19. When patients need to have emergency surgery but have no relevant test reports, a negative pressure operating room with the air conditioning system off is used. The medical personnel wear effective personal protection both before and after surgery. The medical equipment used for patients with suspected or confirmed COVID-19 infections is then soaked with chlorine disinfectant for at least 30 min and sealed in marked plastic bags before being transported to the disinfection supply center for centralized treatment [7]. After the operation, air disinfection as well disinfection of the surfaces and floor in the operating room are carried out. Postoperative medical waste is marked for proper disposal. Some small operating rooms or outpatient examination rooms, such as the digital substraction angiography (DSA) room, dialysis room, and induced abortion room, are set up separately for regular patients and for patients with COVID-19 infections.

## 7. Classifying Areas by Risk of Exposure

The district management of hospitals was a necessary routinely measure implemented in consideration of the pandemic. The entire hospital was divided into areas classified into four grades of risk of exposure with different sets of recommended safety measures according to the degree of viral exposure (Table 1). After the hospital resumed normal operation, the rating system remained alongside environmental monitoring and the classification of different levels of medical protection required. Areas with low risk of exposure include general outpatient clinics, general wards, observation wards, the medical technology department, the administrative/financial/work office, the dining room, and toilet rooms. Areas with medium risk of exposure include respiratory outpatient clinics, hemodialysis centers, disinfection supply centers, injection centers, puncture centers, various surgeries and interventional treatment centers, or areas where patients can remove their masks. The safety measures for areas with low and medium risk mainly include the wearing of work clothes and medical surgery masks. Areas with high risk of exposure consist of the emergency department, abnormal patient observation wards, outpatient departments of stomatology, outpatient departments of otolaryngology and the laryngoscopy room, digestive endoscopy centers, and respiratory endoscopy centers. In these areas, besides the two basic measures mentioned above, isolation clothes and safety goggles should be worn if necessary. Areas with extra-high risk of exposure include fever outpatient clinics; the PCR laboratory; and some operation rooms performing operations that may produce aerosols such as endotracheal intubation and related operations including cardiopulmonary resuscitation (CPR), bronchoscopy, sputum suction, pharyngeal swab sampling, etc. Safety measures for these areas include wearing work clothes, medical protective masks, disposable operating caps, disposable latex gloves, isolation clothes, and safety goggles.

## 8. Protecting Medical Personnel

All medical staff are required to pay attention to the following three protection requirements [8]: first, to understand standard preventive measures; second, to wear work clothes and disposable surgical masks correctly, and if necessary, to wear isolation clothes and goggles/protective face masks; and third, to pay attention to hand hygiene when removing personal protective equipment. All of the clinical medical personnel, medical technicians, cleaners, security personnel, and other administrative personnel should receive regular training about COVID-19 infection prevention. Everyone in the hospital needs to understand the importance of disinfecting their surroundings such as the air, surfaces, ground, and medical equipment and of paying more attention to medical waste disposal.

Occupational exposures to COVID-19 typically result from skin infections, eye conjunctiva infections, respiratory infections, and medical sharp instrument injuries, which should be reported immediately to the hospital’s infection department [9]. Following, doctors should evaluate their conditions and suggest whether a 14-day medical observation is needed. If one has suspected symptoms of COVID-19 during the observation period, immediate treatment is required.

As for inpatients infected with COVID-19 of an unknown source, several necessary measures must be taken immediately: for example, isolate patients with COVID-19 infections and their close contacts immediately, carry out strict environmental cleaning and disinfection measures, and perform a census amongst other inpatients and hospital staff to avoid further spread. When contacting patients with confirmed or suspected COVID-19 infections, medical staff need to take precautions for personal protection, such as wearing isolation clothes, goggles, and N95 masks as well as strengthening hand hygiene.

After the death of patients with suspected or confirmed COVID-19 infections, the movement of each body is handled in a timely manner by the trained staff under sufficient protection. Patients’ personal belongings cannot be taken home until they have been disinfected. The body is also sent directly to a designated place for cremation by a special vehicle as soon as possible.

## 9. Conclusions

All of these measures were set in place both to curb the pandemic and to meet the needs of patients, which required time and effort. Fortunately, after five months of regular function, no positive case was found among a total of 41,219 people (including hospital staff, patients, and caregivers) tested as of 16 September 2020 for the SARS-CoV-2 nucleic acid in our hospital. We are aware that there are still some limitations to our findings, one of which is that our report has no negative control. In addition, our review only took into account the experience of WRCH in safely resuming hospital function but lacks a comparison with other hospitals. However, as one of the first hospitals designated to addressing the pandemic in China, WRCH has accumulated valuable experience in anti-epidemic processes. The strict hospital reform strategies have been extended nationally, and appropriate changes have been made according to different epidemic situations, which has achieved excellent outcomes during the pandemic.

Safely resuming hospital function must become a primary focus in healthcare systems. Therefore, we shared our experience in Wuhan, China, as an early reference, to better help hospital managements transform a designated COVID-19 facility back into a regularly functioning general hospital. The precautions and procedures can be concluded into two aspects: grading the risk of exposure in areas and redefining functional departments. Resuming function in a hospital should be based on both improvements in quality and guarantees of safety.

## Figures and Tables

**Figure 1 pathogens-11-00452-f001:**
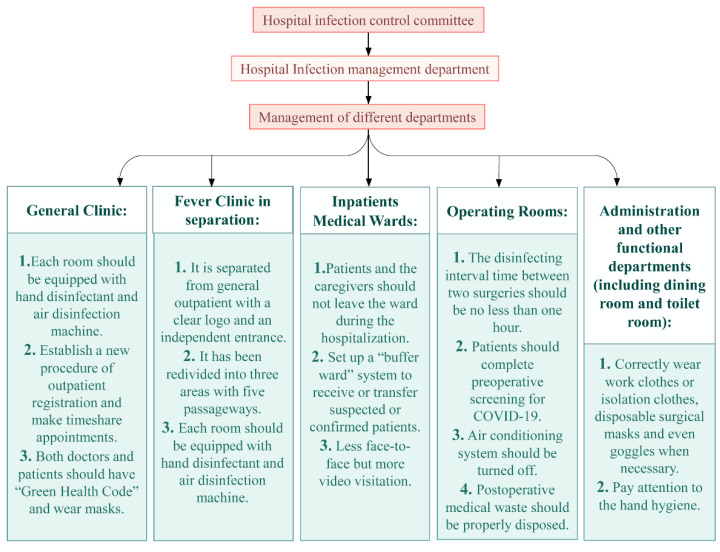
Safety measures taken by a regularly functioning hospital after transformation back from a COVID-19-only care facility (new steps are in italics, compared to before COVID-19).

**Table 1 pathogens-11-00452-t001:** Four-level risk classification of hospital departments and relevant preventive measures for both medical personnel and patients.

Risk Rank	Departments and Areas	Preventive Measures
Low risk of exposure	General outpatient clinic, general ward, observation ward, medical technology department, administrative/financial work office, dining room, toilet room, etc.	1. Work clothes 2. Medical surgical masks
Medium risk of exposure	Respiratory outpatient clinics, hemodialysis centers, disinfection supply centers, injection centers, puncture centers, interventional treatment centers, etc.	1. Work clothes 2. Medical surgical masks
High risk of exposure	Emergency department, stomatology outpatient, otolaryngology outpatient and laryngoscopy room, digestive endoscopy center, respiratory endoscopy center, etc.	1. Work clothes 2. Medical protective masks 3. Isolation clothes 4. Safety goggles (if necessary)
Extra-high risk of exposure	Fever outpatient clinic; PCR laboratory; some operations that may produce the aerosol such as endotracheal intubation, bronchoscopy, sputum suction, pharyngeal swab sampling, etc.	1. Work clothes 2. Disposable operating caps 3. Medical protective masks 4. Isolation clothes 5. Safety goggles 6. Disposable latex gloves

## Data Availability

Not applicable.

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
