# Peer review of "Prevention and Control of COVID-19 after Resuming General Hospital Functions"

_pathogens, 2022, doi:10.3390/pathogens11040452_

Round 1
Reviewer 1 Report
This paper describes the infection control procedures put in place in a hospital that resumed normal operations subsequent to having been repurposed as a COVID-19 treatment facility. The procedures described are common sense precautions based upon our knowledge of transmission of COVID-19, and the hospital has been successful in preventing any further infections.
What is lacking in the introduction is a description of modes of transmission, and the relative importance of the different modes of transmission. One of the reference cited (ref. 2, line 42) addresses this to some extent for SARS-CoV-2 (although the citation is unrelated to the text and could be deleted). I suggest adding the following two sentences at the end of the Introduction, on line 57.
---
"Transmission of SARS-CoV-2 is primarily airborne [4a]. Another route of infection via fomites was believed to be significant early in the pandemic, but has subsequently been shown to be of minor importance [4b]. These principles guided our infection control procedures."
Reference numbering will need to be changed to fit into the overall manuscript. Here are the citations:
4a. Prather KA, Marr LC, Schooley RT, McDiarmid MA, Wilson ME, Milton DK. Airborne transmission of SARS-CoV-2. Science. 2020 Oct 16;370(6514):303-304. doi: 10.1126/science.abf0521.
4b. Goldman E. SARS Wars: the Fomites Strike Back. Appl Environ Microbiol. 2021 Jun 11;87(13):e0065321. doi: 10.1128/AEM.00653-21.
---
My only other comment concerns Figure 1: 2nd column from left has top cut off. Point 1 is truncated from the top of the column "Fever Clinic in separation". This needs to be fixed by the authors.
Reviewer 2 Report
The most overwhelming threat to our overall global health and well-being is the COVID-19 pandemic that we’ve been facing since its discovery in Wuhan, China in late 2019. The protective measures to prevent the spread of COVID-19 in public spaces, especially in hospitals, is an important issue. The authors shared their experience in hospital’s functional resumption by setting up a three-level hospital infection management system and summarized four grades of risk exposure areas, as well as the experience in safety measures of medical personnel in different departments. The manuscript is well written and the experience will be helpful for the establishment of disinfection in hospital. However, it would be better before publication if the following points would be addressed.
- If inpatients were infected with COVID-19 of unknown source, could you please present the management protocol to protect both patients and staffs that may be taken?
- Some outpatients need emergency surgery, such as pregnant women suddenly have to give birth while waiting for the screening results of COVID-19. Could you please give the reporting procedure from the detection staffs to hospital presidents to ensure the pregnant women have surgery in time?
Author Response
The most overwhelming threat to our overall global health and well-being is the COVID-19 pandemic that we’ve been facing since its discovery in Wuhan, China in late 2019. The protective measures to prevent the spread of COVID-19 in public spaces, especially in hospitals, is an important issue. The authors shared their experience in hospital’s functional resumption by setting up a three-level hospital infection management system and summarized four grades of risk exposure areas, as well as the experience in safety measures of medical personnel in different departments. The manuscript is well written and the experience will be helpful for the establishment of disinfection in hospital. However, it would be better before publication if the following points would be addressed.
1. If inpatients were infected with COVID-19 of unknown source, could you please present the management protocol to protect both patients and staffs that may be taken?
Thanks for your insightful question. As for inpatients are found to be infected with COVID-19 of unknown source, several necessary measures must be taken immediately. For instance, isolate COVID-19 patients and their close contacts, and carry out strict environmental cleaning and disinfection measures, especially in areas where infected patients have moved. In addition, do a census among other inpatients and hospital staff to avoid further spread. When contacting with confirmed or suspected COVID-19 patients, medical staff need to take personal protection, such as wearing isolation clothes, goggles and N95 masks, as well as strengthening hand hygiene. And we added the sentences in lines 206-212.
- Some outpatients need emergency surgery, such as pregnant women suddenly have to give birth while waiting for the screening results of COVID-19. Could you please give the reporting procedure from the detection staffs to hospital presidents to ensure the pregnant women have surgery in time?
We greatly appreciate your excellent suggestions. Actually, WRCH (Wuhan Red Cross Hospital) had formulated an emergency plan for outpatients who need emergency surgery but failed to provide the screening results of COVID-19. Specifically, those patients were allowed to be admitted into the hospital without contact and transferred to a dedicated operation room for timely treatment. Meanwhile, detection staff would conduct rapid screening for them. And we added the sentences in the revised manuscript (lines 116-120).
Reviewer 3 Report
Authors gave done a great work to share their experience for infection prevention a their hospital regarding COVID-19. It is being presented as a review. In the abstract "After five months of regular function, more than 40 thousand people have been tested so far without a single positive case found in our hospital." They should clarify whether these "people" include only the hospital staff or also patients and their caregivers.
Introduction is well written and highlights the importance of this article as several healthcare system scrambled to address this issue. Their approach is clearly defined in the text, figures and tables are appropriately used. They have identified the limitations such as no control to compare their success rate. Unclear what the authors meant by the following statement in the conclusion "Secondly, the pandemic shall disappear eventually and we shall know these preventive measures at that time."
They should also discuss whether these measures were universal in China during the pandemic or different hospital adopted different approach.
They should also consider adding timeline as it has been couple of years since the pandemic started, so that readers can understand the context of this study better.
Author Response
- Authors gave done a great work to share their experience for infection prevention a their hospital regarding COVID-19. It is being presented as a review. In the abstract "After five months of regular function, more than 40 thousand people have been tested so far without a single positive case found in our hospital." They should clarify whether these "people" include only the hospital staff or also patients and their caregivers.
Thanks for your careful review and we rewrote the sentence as “After five months of regular function from April to September in 2020, more than 40 thousand people, including hospital staff, patients and caregivers for patients, had been tested without a single positive case found in our hospital after all patients discharged.” (lines 24-25).
- Introduction is well written and highlights the importance of this article as several healthcare system scrambled to address this issue. Their approach is clearly defined in the text, figures and tables are appropriately used. They have identified the limitations such as no control to compare their success rate. Unclear what the authors meant by the following statement in the conclusion "Secondly, the pandemic shall disappear eventually and we shall know these preventive measures at that time."
Thank you and we rewrote the sentence as “In addition, our review only concluded the experience of WRCH in safe hospital resumption, but lack the comparison of other hospitals’ measures” in the revised manuscript (lines 222-224).
- They should also discuss whether these measures were universal in China during the pandemic or different hospital adopted different approach.
Thanks for your insightful suggestion. We have put this point as a limitation. We added the sentences “In addition, our review only concluded the experience of WRCH in safe hospital resumption, but lack the comparison of other hospitals’ measures. However, as one of the first designated hospitals in China, WRCH has accumulated valuable experience in anti-epidemic process. These strict hospital reform strategies have been extended nationally and made appropriate changes according to different epidemic situations, which has achieved excellent outcomes during the pandemic.” in the revised manuscript(lines 223-229).
- They should also consider adding timeline as it has been couple of years since the pandemic started, so that readers can understand the context of this study better.
Thanks for your advice and we have added timeline as shown in lines 24-25(After five months of regular function from April to September in 2020), line 30(three years), line 38(in late March,2020), line 43 (in April 2020), line 54(three months from January to April 2020), line 219(after five months of regular function) and line 221 (September 16, 2020) in the revised manuscript.